# Variation in Rice Plastid Genomes in Wide Crossing Reveals Dynamic Nucleo–Cytoplasmic Interaction

**DOI:** 10.3390/genes14071411

**Published:** 2023-07-08

**Authors:** Weilong Yang, Jianing Zou, Jiajia Wang, Nengwu Li, Xiaoyun Luo, Xiaofen Jiang, Shaoqing Li

**Affiliations:** 1State Key Laboratory of Hybrid Rice, Hongshan Laboratory of Hubei Province, Key Laboratory for Research and Utilization of Heterosis in Indica Rice of Ministry of Agriculture, Engineering Research Center for Plant Biotechnology and Germplasm Utilization of Ministry of Education, College of Life Science, Wuhan University, Wuhan 430072, China; yangweilong@whu.edu.cn (W.Y.); 2018202040067@whu.edu.cn (J.Z.); 2016282040193@whu.edu.cn (J.W.); qwe8341110@163.com (N.L.); 2015202040067@whu.edu.cn (X.L.); 15036696377@163.com (X.J.); 2Institute of Advanced Biotechnology and School of Life Sciences, Southern University of Science and Technology, Shenzhen 518036, China

**Keywords:** plastid, rice, repeat sequence, RNA editing, nucleo–cytoplasmic interaction

## Abstract

Plastid genomes (plastomes) of angiosperms are well known for their relative stability in size, structure, and gene content. However, little is known about their heredity and variations in wide crossing. To such an end, the plastomes of five representative rice backcross inbred lines (BILs) developed from crosses of *O. glaberrima*/*O. sativa* were analyzed. We found that the size of all plastomes was about 134,580 bp, with a quadripartite structure that included a pair of inverted repeat (IR) regions, a small single-copy (SSC) region and a large single-copy (LSC) region. They contained 76 protein genes, 4 rRNA genes, and 30 tRNA genes. Although their size, structure, and gene content were stable, repeat-mediated recombination, gene expression, and RNA editing were extensively changed between the maternal line and the BILs. These novel discoveries demonstrate that wide crossing causes not only nuclear genomic recombination, but also plastome variation in plants, and that the plastome plays a critical role in coordinating the nuclear–cytoplasmic interaction.

## 1. Introduction

The plastid is a vital organelle in plant cells, and is the main location of photosynthesis, where starch, pigments, fatty acids, and other important compounds are synthesized [1]. Unlike the nuclear genome, plastids mainly show uniparental inheritance in plants [2,3]. Specifically, there is maternal transmission of plastids in plants, especially for angiosperms and the gymnosperm groups cycads and gnetophytes [4,5]. Furthermore, the plastid genomes (plastomes) in land plants usually have a stable scale ranging from 75 to 200 kb and highly conserved structures and organization of content [6]. Their structure consists of a single circular molecule with a quadripartite structure, which includes two copies of an inverted repeat (IR) region that separates the large and small single-copy (LSC and SSC) regions [7]. Similar to mitochondrial genomes (mitogenomes), many repeated sequences have also been found in plastomes [8]. Although the structure of plastomes is more stable than mitogenomes, which may be due to the IRs, genome rearrangements have also been found in plastomes, such as *Campanulaceae* [9], *Fabaceae* [10], *Geraniaceae* [11], and *Oleaceae* [12]. In reality, homologous recombination between repeat sequences and IR boundary shifts (expansions and contractions) is the key factor which causes genome rearrangements in plastomes [13]. Compared with mitochondria, the variation in plastomes is not so prevalent in plants; however, it has great significance in genome diversity, environmental adaptation, plant growth and development, etc.

During the long process of evolution, massive genes in the ancestor of the plastid were transferred to the nucleus [14]. Plastid physiological activities depend on the coordinated interaction between nuclear-encoded proteins and their counterparts produced in the plastomes. Consequently, the genes in nuclear genomes and plastomes are thought to be coadapted, ensuring normal physiological activity in plants [15]. The functional interactions between nucleo–cytoplasmic genomes have vital evolutionary significance [16]. The incongruous nuclear and cytoplasmic genotypes created by hybridization often have remarkable phenotypic effects, sparking a recent controversy about whether nucleo–cytoplasmic interactions lead disproportionately to speciation [17,18,19,20]. Importantly, nucleo–cytoplasmic interactions also have influences on the growth, development, and even survival of plants [21,22]. Although the coordinated nucleo–cytoplasmic interaction is very significant, the underlying mechanism remains poorly understood.

Inter- or intraspecies crossing will lead to a reshuffle of the nuclear genome and hence renewed cytonuclear combination. Thus, how to keep a coordinated mitochondrial–nuclear interaction will be closely related to the normal growth and development of the plants. We speculated that the mitogenome may have evolved some mechanisms to cope with the reshuffle of nuclear genomes. In order to answer this hypothesis, we constructed a BC_2_F_12_ backcross inbred line (BIL) population in a previous study whose organellar genomes were all inherited from the same maternal parent using crosses of *O*. *glaberrima* × 9311 [23]. We investigated the mitogenomes of the stable BILs and found that the structure and organization of mitogenomes, gene copy number, expression, and RNA editing were changed among the BILs. However, less information is known about plastomes in BILs. Thus, herein, the plastomes of five representative BILs were constructed and the genomic structure, gene content, repeat sequences, gene expression, and RNA editing were compared between the maternal line (*O*. *glaberrima*) and the BILs with diverse genotypes. Our study provides hard proof that plastomes will change to adapt to the reshaped nuclear genome for cytonuclear coordination when wide crossing happens.

## 2. Materials and Methods

### 2.1. Plant Materials

Five representative BILs (BC_2_F_12_), P10, P88, P90, P91, and P92, derived from crosses of *O. glaberrima*/*O. sativa*, and the maternal parent line *O. glaberrima*, were used in the paper. The planting and management of the rice lines followed the methods in our previous study [23]. 

### 2.2. Plastomes Assembly and Annotation

Firstly, we used raw reads, which were longer than 26 Kb, from PacBio RSII as seed reads to construct original plastomes. Next, the raw reads shorter than 26 Kb were used to correct original plastomes by the Pacific Biosciences SMRT analysis (v2.3.0) software package with default settings. In addition, the plastomes were also corrected by the raw reads from Illumina HiSeq Xten by pilon (--changes--vcf--fix bases--threads 5--mindepth 10). DOGMA [24] was used to annotate plastid genomes. The tRNA genes in plastomes were detected by tRNAscan-SE [25]. Finally, the plastid genome maps were illustrated by OGDRAW [26]. 

### 2.3. Detecting Genome Rearrangement

To accurately detect the rearrangement events in plastomes, one IR region was moved from the plastomes. Next, the rearrangement events of the plastomes were detected by Mauve Version 2.3.1 [27] and all plastomes were aligned with an LCB cutoff of 500.

### 2.4. Repeats, Repeat-Mediated Recombination, and Molecular Verification

Repeats were analyzed by BLASTN to search the *O. glaberrima* plastome against itself with a word size of 20 nucleotides and an expectation value of 1. We counted the copy numbers of each repeat and distinguished them based on the flanking sequences [28]. The repeat sequences with only two copies were selected to design primers (Appendix A) by Primer 5 [29] to analyze repeat-mediated recombination.

The procedures devised by Mower et al. [30] were used to detect the frequency of repeat-mediated recombination in plastomes and we used the calculation methods and parameters reported by Guo et al. [31] for this analysis. Moreover, we calculated the content of one repeat configuration as the mapping read pair number divided by all mapping read pairs of repeat configurations, and we calculated the relative content of one repeat configuration in BILs as the content of repeat configuration in BILs divided by the content of repeat configuration in *O. glaberrima*.

For verification, total DNA was isolated using the CTAB method [32] from the 2-week-old rice seedlings. Real-time quantitative PCR was used to verify repeat-mediated recombination by LightCycler 480 (Roche) and the SYBR Green I Master PCR kit (Roche).

### 2.5. Transcriptome Analysis and RNA Extraction

The procedures devised by Guo et al. [33] were performed to trim adapter and low-quality reads of RNA sequencing data. Clean reads were mapped to plastomes by Bowtie2 v2.3.4.1 [34] (--no-mixed--no-discordant--gbar 1000--end-to-end-k 200-q-X 800) and the FPKM of the gene was calculated by RSEM v1.2.25 [35].

Total RNA of the 2-week-old rice seedlings was extracted using TRIzol reagent (Invitrogen). Approximately 4 μg of RNA was treated with DNase I (NEB) and reverse transcripted by SuperScriptII to obtain cDNA using the experimental procedure in the manufacturer’s instructions. Real-time quantitative PCR was used to verify repeat-mediated recombination by LightCycler 480 (Roche) and the SYBR Green I Master PCR kit (Roche) to verify transcriptome analysis.

### 2.6. RNA Editing Analysis

RNA editing sites were detected using the pipeline of Guo et al. [33] and the editing sites having a cover depth ≥ 50 were kept using SAM tools mpileup [35]. For verification, the cDNA with the editing site was obtained and ligated into pMD18-T vector and the products were co-transformed into *E. coli*. Subsequently, 50 monoclonal constructs of each editing site were picked up to conduct Sanger sequencing and evaluate the RNA editing rate.

## 3. Results

### 3.1. Organization of O. glaberrima and BIL Plastomes

To construct plastomes of *O. glaberrima* and five BC_2_F_12_ BILs (P10, P88, P90, P91, and P92) with different nuclear genomes, we recalled the sequence data in our previous study (Appendix A). Each plastome was assembled into a circle, with stable GC content (39%), and genome size varied from 134,540 to 134,598 bp (Figure 1 and Table 1). All plastome constructions had a conservative quadripartite structure of angiosperms, containing two inverted repeats (IRs) whose size ranged from 20,803 bp to 20,804 bp, separated by a pair of single-copy regions: one was a large single copy (LSC, ranging from 80,551 bp to 80,609 bp) and the other was a small single copy (SSC, ranging from 12,381 bp to 12,382 bp).

The plastomes’ coding regions were relatively stable, ranging from 87,804 bp to 87,822 bp; however, the non-coding regions changed, ranging from 46,736 bp to 46,776 bp (Table 1), suggesting the non-coding regions were the major contributor to the change in BIL plastomes. After annotation, we found each BIL plastome had the same sets of protein, rRNA, and tRNA genes, which were the same as the maternal line (*O. glaberrima*). These included 76 protein genes, 4 rRNA genes, and 30 tRNA genes. Among them, fifteen genes, *atpF*, *ndhA*, *ndhB*, *petB*, *petD*, *rpl16*, *rpl2*, *rps12*, *rps16*, *trnA-UGC*, *trnG-UCC*, *trnI-GAU*, *trnK-UUU*, *trnL-UAA*, and *trnV-UAC*, contained only one intron; one gene (*ycf3*) contained two introns; eighteen genes had two copies; and only one gene had three copies.

### 3.2. Plastomes Recombination and Variation

All plastomes of *O. glaberrima* and the five BILs were compared; however, a genome rearrangement event was not found (Appendix A). Although the plastomes of the BILs were stable, the recombination rate of repeats in the plastomes was suspected of changing, similar to that in mitogenomes. Therefore, we explored the repeats (>20 bp) for their type and distribution, which are considered the main causes of the rearrangement in plants’ plastomes (only one IR was contained). Ultimately, 16 repeats were found in the *O. glaberrima* plastome using BLASTALL, which covered 0.52% of the plastome (Appendix A). According to the repeats and flanking sequences, there were eight different types of repeats and all of them had a pair of copies (Appendix A).

The recombination of two-copy repeats generates only two products, which makes them convenient for tracing (Figure 2A). Therefore, the reads that came from the 20 Kb PacBio sequencing library were mapped to the plastid repeat sequences, and obvious changes in the repeat configurations (repeat + flanking sequence) belonging to the same repeat were found among BILs. The results were shown by a reduction in CRS8-cd to 38.1% in P92 and an increase in MRS3-ad to 197.8% in P92 compared to *O. glaberrima* (Appendix A). Interestingly, homologous recombination (repeat-mediated) was found in BILs to a great extent (Figure 2B), which was further demonstrated by qPCR showing that both CRS8-cb in the P88 plastome and CRS3-cb in the P92 plastome were about 2.5 times higher than that in the maternal line (Appendix A). Thus, the results we found suggest the plastomes were not stable in the BILs.

### 3.3. Expressional Profile of Plastid Genes in BILs

Repeats may control the expression of the gene in both plastids [36] and mitochondria [37], which has been reported in previous studies. In BIL mitogenomes, the gene expression was influenced by repeat recombination [23]; therefore, we wondered if the same thing happened in plastomes. Our analysis showed that fewer repeats were found in plastomes compared to mitogenomes; however, almost all repeats were distributed in the regions located 2 Kb upstream or downstream from genes (Appendix A). Transcriptome analysis was performed to analyze the expression profile of plastid genes, which was also verified by qPCR. The results indicate that the expression level of functional genes in plastids among the BILs ranged from 0.12 to 2.87 times compared to the maternal line (Figure 3A), which had less variation than mitochondrial genes (0.27 to 4.16). In addition, the expression of *psbZ*, *psbK*, *rbcL*, *psbH*, *ndhD*, and *psbE* decreased by more than 50% compared to the maternal line; however, the expression of 10 genes (*atpF*, *rpl33*, *rpl20*, *rps8*, *rps2*, *rpl22*, *rpl23*, *rps14*, *ccsA*, and *rps3*) increased by more than 50% compared to *O. glaberrima* (Figure 3A). Furthermore, we found that 50% of the genes that were more than 50% down-regulated and 10% of the genes that were more than 50% up-regulated were all located within 2 Kb of the repeats. Finally, there was no obvious positive correlation between expression levels and their gene contents, based on qPCR analysis (Appendix A), suggesting a vital role of repeats in regulating plastid gene expression, which was also found in mitogenomes.

### 3.4. RNA Editing Pattern of Plastid Genes in BILs

A previous study showed that RNA editing contributes to coordinating plastid–nuclear interaction [38]. Hence, we explored the RNA editing sites and rate of all coding genes with reads covered at least 50 times in every plastome (Table 2 and Appendix A). In total, 52 C-to-U and 22 U-to-C editing sites located in 32 plastid genes were detected in the protein-coding regions, which was far less than that in mitogenomes (525 C-to-U and 4 U-to-C editing sites) (Appendix A). Furthermore, about 50% of the editing sites were observed at the second codon, and 51 editing sites contributed to amino acid variation (Appendix A) in 32 genes (Appendix A).

Compared to *O*. *glaberrima*, the turnover of the RNA editing rate in the same position was frequently found among BIL plastids (Figure 3B), which was the same as mitogenomes [23]. For instance, the site of *ndhC*-358 had an editing rate of only 17.2% in *O. glaberrima*, but it was 6.3% in P88 and 25.3% in P91 (Figure 3B), showing that the RNA editing was dynamic even in the same position among different lines. To certify the veracity of the bioinformatics results, three RNA editing sites with abundant variation (*matK*-1351, *ndhA*-1070, and *rpoB*-545) were chosen for re-sequencing, and all editing sites had semblable editing patterns to bioinformatics results (Appendix A), suggesting the uniformity between them.

## 4. Discussion

Traditionally, plastomes were considered stable, displaying hardly any change. Nevertheless, in this study, variations in the rice plastid genomes from the same maternal parent were observed, revealing a new way for the plastome to coordinate with a reshuffled heterogenic nuclear genome. When compared to mitogenomes in BILs, the plastomes were relatively stable. However, extensive differences were observed in repeat-mediated recombination, gene expression, and RNA editing, indicating that the inheritance of plastomes across the descendent plants was not constant, and each progeny had specific plastid genomic types. Similar to mitochondria, there was not always a positive correlation between expression levels and copy number in detected BILs, suggesting that plastids coordinate nuclear–cytoplasmic interaction in a complex way.

### 4.1. Plastids and Mitochondria Show Distinctively Different Genomic Organization 

Unlike mitogenomes, no obvious genomic rearrangement was found among the maternal and BIL plastomes (Appendix A); however, a few repeat-mediated recombinations, which were the main contributor to genomic rearrangements [39,40], were detected in BIL plastomes (Figure 2B). What caused this phenomenon to happen? There are three main reasons: (1) Unlike the diverse structure of the mitogenome, the quadripartite structure is typical for the plastome, owing to two copies of an IR region that may work as a vital contributor to plastome stability among plants [13]. (2) There are more repeats in mitogenomes than in plastomes. In *O. glaberrima*, 118 (≥50 bp) repeats were found in the mitogenome [23], yet only 16 (>20 bp) repeats were found in the plastome (Appendix A). (3) Plastomes (only one IR was contained) contain no large repeat (>1000 bp), since large repeats in the mitogenome will lead to high-frequency DNA recombination, contributing to the fission of the genome to a series of multipartite configurations [31,41], as shown in *O. glaberrima* [23]. In general, the number and length of repeats in the genome are the key reasons why plastids and mitochondria exhibit distinctively different genomic organization.

Mitochondria and plastids have many similar characteristics, such as a relatively independent genome, uniparental inheritance, gene transfer to the nuclear genome, a multi-copy genome, the ability to form complexes co-produced with the nuclear genome, and so on. However, varying degrees of change between the mitogenome and plastome in BILs magnify the difference in inheritance between mitochondria and plastids. Indeed, the mitogenome has been reported to change by repeat-mediated recombination in many different kinds of plants, even some new *orf*s were produced by mitogenomic recombination in some special cases [28]. Interestingly, some newly derived *orf*s even influence plants’ physiological activities, exemplified by cytoplasmic male sterility (CMS) [42,43]. When it comes to the plastome, hardly any information about new *orf*s produced by DNA recombination has been reported. Although they are both organelles, they play different roles in genomic variation: mitochondria are “activists” and plastids are “idlers”. Although plastids are not very active, variations were detected in this study. Therefore, more effort is needed to study chloroplast genome variation, uncovering the mechanisms and evolutionary implications of this process.

### 4.2. Plastids and Mitochondria Have Different RNA Editing Patterns

Contrary to mitogenomes, no gene copy number and gene order were changed in BIL plastomes. However, the expression of some genes changed extensively (Figure 3A), which may be due to repeat-mediated recombination and nuclear genome reprogramming. In turn, the various transcriptomes of plastomes may contribute to adapting the reshuffled heterogenic nuclear genome. Our findings imply that the dosage effect has a significant contribution in regulating plastid–nuclear interaction; however, the related mechanism is still unclear.

Growing evidence proves that RNA editing plays an important role in regulating plant growth and development [44]. Significantly, RNA editing is thought to participate in the rescue of organelle dysfunction generated by genetic, physiological, or environmental factors [38]. In our research, 74 RNA editing sites were detected in plastomes, which was far less than that in mitogenomes (529) [23]. As we all know, the base substitution rate in mitogenomes is slower than that in plastomes and nuclear genomes in seed plants [45]. Unlike in animals, RNA editing has been widely detected in plastomes and mitogenomes of plants [46], which means amino acid sequences are determined by both DNA sequences and RNA editing. Therefore, more RNA editing, to some extent, can compensate for the slower substitution rate in plastomes and mitogenomes, adapting to a faster substitution rate in nuclear genomes. 

More importantly, in this study, about 18.9% of the RNA editing sites showed editing rate destabilization among the BILs compared to *O. glaberrima* (Figure 3B). The changeability in the RNA editing site and rate in the BILs is strongly consistent with the nuclear heterogeneity, reflecting the co-adaption between the plastome and nuclear genome to some extent. Furthermore, the same phenomenon was also detected in mitochondria [23]. Although the RNA editing compensatory effect on the substitution rate is an important part of coordinating the nuclear–cytoplasmic interaction, it has always been grossly overlooked in previous studies. Researchers have studied nuclear–cytoplasmic interaction at the DNA level, which misses the contribution of RNA editing. Therefore, the relevant research performed at the RNA level may allow us to uncover more interesting results.

### 4.3. Wide Crossing Contributes to the Diversity of Cytoplasmic Genomes

Natural and artificial out-crossing is easily found in plants [47], especially in commercial crops such as rice [48], cotton [49], rape seed [50], sorghum [51], etc. Generally, the wide crossing should only change the nuclear genome, since cytoplasm in the bulk of the higher plants is usually maternally inherited. Nonetheless, we found that the repeat-mediated recombination, gene expression, and RNA editing in mitochondria and plastids were extensively altered between the maternal line and the BILs, indicating that wide crossing could also create cytoplasmic genome diversity, which is also an important part of plant genome diversity. 

To explain how this alteration occurs, we propose that some proteins, encoded by nuclear genes, are transferred to mitochondria and plastids to regulate their physiological and biochemical activity [16], especially the genes involved in DNA replication, recombination, and repair (RRR) [52]. These have been reported to regulate the recombination of repeat sequences of the cytoplasmic genomes, such as *MSH1* [53], *RecA3* [54], *RecG1* [55], *OSB1* [56], and *Why2* [57]. Recently, MSH1 has been proven to recognize and correct errors in plant mitochondria and plastid DNA sequences [58]. Additionally, the pentatricopeptide repeat (PPR) gene family widely participates in the RNA editing in plastomes and mitogenomes to regulate the cytoplasm to adapt to the nucleus [59,60]. The wide crossing will lead to a reshuffle of the nuclear genome and hence the change in expression and gene combinations from different parents. In this process, some nuclear genes will modify mitochondria and plastids to a form which can interact harmoniously with the nuclear genome, resulting in the normal growth and development of the plants. Of course, there are many relevant nuclear genes that are unknown, which requires us to explore further.

## 5. Conclusions

The plastomes of five representative rice backcross inbred lines, which were made by crosses of *O*. *glaberrima*/*O*. *sativa*, were systematically analyzed in this study. These displayed a stable size, structure, and gene content; however, repeat-mediated recombination, gene expression, and RNA editing were dynamic. Furthermore, our molecular experiments provided solid evidence for the existence of these phenomena. This finding not only expands our understanding of how plant chloroplast genomes are inherited, but also improves our understanding of the coordinated nucleo–cytoplasmic interaction mechanism.

## Figures and Tables

**Figure 1 genes-14-01411-f001:**
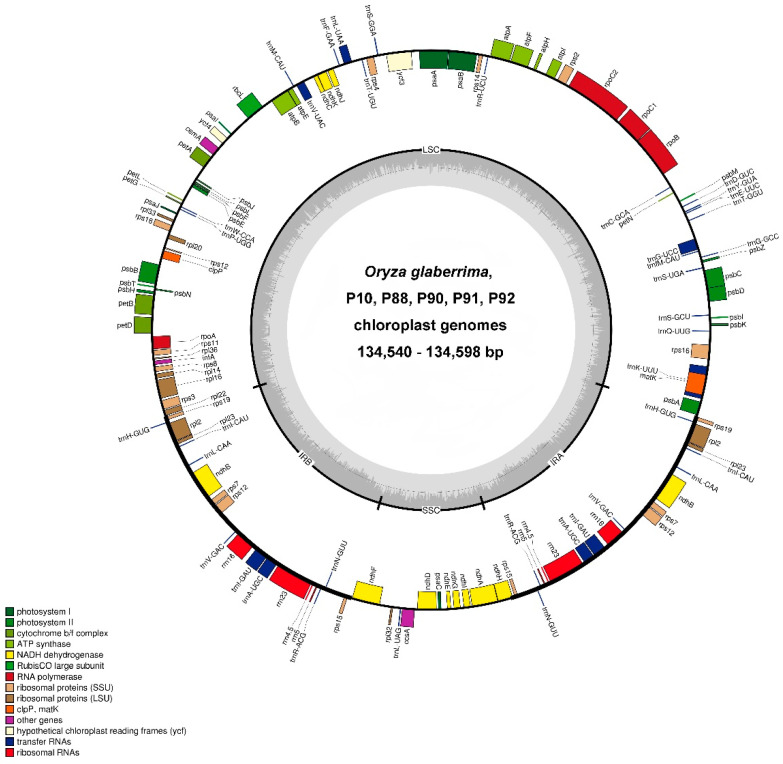
Complete plastomes of the maternal parent (*O. glaberrima*) and five BILs. The genes drawn outside of the circle are transcribed counterclockwise, while those inside are clockwise. Two inverted repeat sequences (Ira and IRb), large single copy (LSC) and small single copy (SSC), are marked. GC content of the genome is also indicated. The inner legend displays the gene function or identifiers using colors.

**Figure 2 genes-14-01411-f002:**
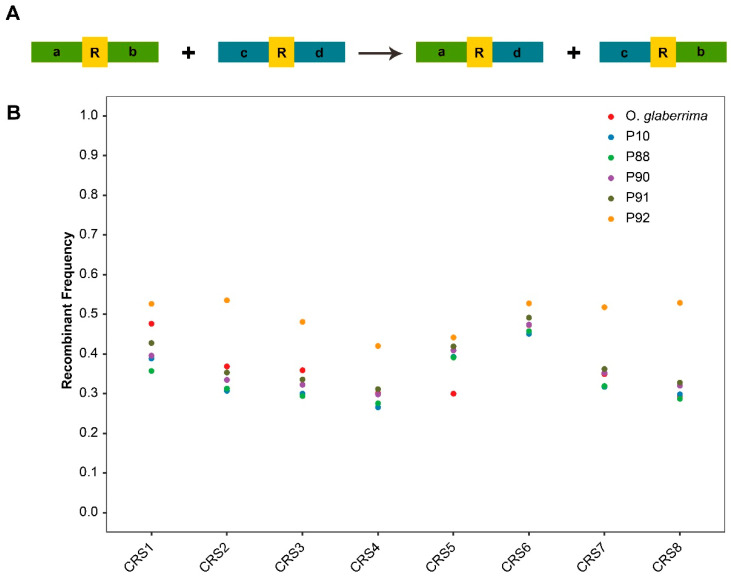
The repeat recombination in plastomes of BILs. (**A**) The diagram shows the recombination manners of two-copy repeats. (**B**) Recombination rate of all two-copy repeats in BIL plastomes.

**Figure 3 genes-14-01411-f003:**
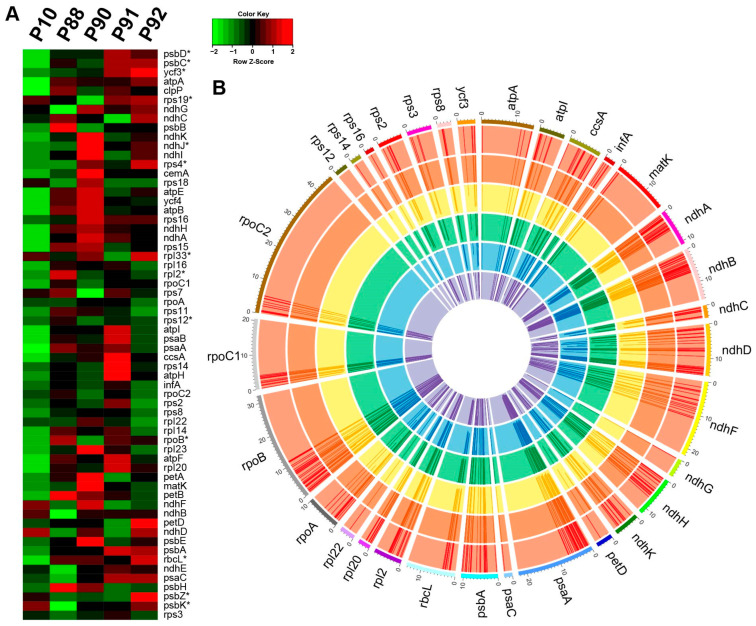
Relative expression and RNA editing rate and sites of plastid genes in *O. glaberrima* and BILs. (**A**) Heatmap of the RNA level of plastid genes. The asterisk marks the gene that is within 2 Kb of repeat. (**B**) Circle map shows the RNA editing rate and sites of plastid genes. From inside to outside, the circles in order represent *O*. *glaberrima*, P10, P88, P90, P91, and P92, respectively. The RNA editing rate is represented by the line length in the circles. The length equal to the circle width represents 100% RNA editing rate and as the line length becomes shorter, the RNA editing rate decreases proportionally.

**Table 1 genes-14-01411-t001:** General information on the plastid genomes in BILs.

Lines	Genome Size (bp)	GC Content (%)	Gene Region Size (bp)	Non-Gene Region Size (bp)	LSC Region Size (bp)	SSC Region Size (bp)	IR Region Size (bp)
*O. glaberrima*	134,561	39.00	87,810	46,751	80,572	12,381	20,804
P10	134,540	39.00	87,804	46,736	80,551	12,381	20,804
P88	134,598	39.00	87,822	46,776	80,609	12,381	20,804
P90	134,575	39.00	87,822	46,753	80,587	12,382	20,803
P91	134,596	39.00	87,822	46,775	80,609	12,381	20,803
P92	134,595	39.00	87,821	46,774	80,606	12,381	20,804

**Table 2 genes-14-01411-t002:** Comparison of the RNA editing in plastomes of the BILs.

Lines	Edited Genes	Edited Sites	Editing-Rate-Changed Sites (vs. *O*. *glaberrima*)
*O*. *glaberrima*	28	65	-
P10	27	62	10
P88	27	63	11
P90	28	66	9
P91	30	68	14
P92	28	65	12

Notes: A site where the editing rate increased or decreased more than 10% is defined as an editing-rate-changed site.

## Data Availability

All PicBio data, NGS data, and RNA-Seq data of plastids are available in the NCBI SRA database (PRJNA598996). All plastomes produced in this study were deposited in GenBank (MW528837-MW528842).

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
