# Peer review of "Variation in Rice Plastid Genomes in Wide Crossing Reveals Dynamic Nucleo–Cytoplasmic Interaction"

_genes, 2023, doi:10.3390/genes14071411_

Round 1

Reviewer 1 Report

There are some journal guidelines that need to be revised especially on the correspondence and author information.

Also, please check journal guidelines with regards to the insertion of figures as they are placed before the mention of Figures 1-3 in the texts. 

In the introduction, it may be better to indicate as to why these inbred lines from O. glaberrima and O. sativa were chosen.  Although it might have been indicated in your previous study, it would be better to reemphasize it in this section. 

Reviewer 2 Report

Presented work on rice plastid genomes is highly significant for peer plant genetics researchers working to advance knowledge about the role of plastomes in gene rearrangement and RNA-editing patterns. The authors have presented their work succinctly; however, the presented result section is not strong enough and I wish the authors would have presented the result findings coherently. The presented manuscript needs further work to be considered in its current form hence authors are recommended to make a major revision for further consideration.

For more detailed and specific comments please see in the review report.

Overall, the use of English is appropriate, and did not notice any major errors or mistakes. However, there are a few typos that need to be addressed.

Round 2

Reviewer 2 Report

Overall authors have taken significant efforts to revise the manuscript and address the review comments; however, there are some issues that remain unaddressed and are still repeated (either there is an error in the submission system or the authors may have missed it somehow). 

The manuscript could be further considered if these issues are addressed accordingly.

English is overall okay, authors can cross-check again for any overlooked grammatical issues if needed.
